# FusionRender: Harnessing WebGPU's Power for Enhanced Graphics Performance on Web Browsers

Submission Id: 465

## ABSTRACT

Graphics rendering on web browsers serves as the foundation for numerous web applications. In contrast to the widely employed WebGL, the next-generation web graphics API, WebGPU, demonstrates an enhanced capacity to adapt to modern GPU features, boasting even more significant potential. Nevertheless, our evaluation shows that the current prevalent performance of graphics rendering frameworks based on WebGPU lags behind those built on WebGL. This discrepancy primarily arises from an incomplete alignment with WebGPU's distinctive attributes. The individual rendering of each graphic leads to redundant communication between the CPU and GPU. To enhance the graphics performance on the web, we introduce the FusionRender to harness the power of WebGPU. To mitigate redundant communication, it assigns a unique signature to each object that requires rendering and employs these signatures for grouping, enabling the consolidation of graphic rendering whenever possible. In simulated experiments involving the rendering of multiple objects, FusionRender demonstrates a median rendering performance improvement of 29.3%-122.1% compared to the existing optimal baseline. In real cases with more complex features, performance improvement ranges from 9.4% to 39.7%. Additionally, FusionRender exhibits robust performance across various devices and browsers.

## CCS CONCEPTS

• **Information systems** → **Web applications**; • **Computing methodologies** → **Rendering**; • **Software and its engineering** → **Software performance**.

## KEYWORDS

Web applications; Graphics; WebGPU; Performance optimization

## 1 INTRODUCTION

Web graphic rendering is foundational for many web applications, encompassing online games [7, 12, 15], web-based virtual reality [2, 17, 34], and digital exhibitions [13, 19]. While WebGL [64] currently stands as the extensively adopted web graphics API, its foundation in OpenGL ES [37] design restricts its compatibility with contemporary hardware owing to historical constraints. Emerging as the next-generation web graphics API, WebGPU [65] is built upon Microsoft's Direct3D 12 [52], Apple's Metal [18], and The Khronos Group's Vulkan [38]. In contrast to WebGL, WebGPU holds more significant potential, promising new avenues for the evolution of graphic rendering in web browsers. Main browsers, including Chrome and Firefox have laid the groundwork for preliminary WebGPU support [50, 67]. Safari used to support WebGPU but has discontinued the experimental feature and is undergoing modifications [5, 51]. In April 2023, WebGPU was enabled by default in the Chrome browser's testing version [21].

Currently, there is rapid development in the field of graphics rendering frameworks utilizing WebGPU. Several previously popular graphics rendering frameworks in web browsers have incorporated support for WebGPU, such as Three.js [14], Babylon.js [3], and PlayCanvas [11]. Besides, novel graphics rendering frameworks based on WebGPU, like Orillusion [9], have also emerged. However, our understanding of their performance remains limited. We have collected information on several prominent frameworks to grasp the current landscape of WebGPU-based graphic rendering frameworks and conducted preliminary performance assessments. However, contrary to expectations, the performance of WebGPU-based rendering falls behind that of WebGL-based frameworks. This discrepancy conflicts with WebGPU's design objectives and our preliminary performance tests.

The main issue here is that the current framework does not effectively harness the potential of WebGPU. It involves separate state configuration and data transmission for each graphic, sequentially rendering different graphics and communicating draw commands and data for each one individually. However, due to WebGPU's capability for global pipeline configuration, the problem encountered in WebGL, where sequentially configuring each state is necessary due to using a global state machine, can be avoided. As a result, multiple graphics rendering processes can be merged to reduce communication overhead.

In pursuit of more efficient web graphic rendering, we aim to maximize the consolidation of graphics rendering to minimize redundant communication overhead. To achieve the objective, we face several challenges. First of all, determining which objects can be consolidated poses a question. The feasibility of merging diverse graphics into a single rendering pipeline hinges on something other than the graphics' shapes but rather on their rendering configurations. Besides, the consolidation of data and computations demands attention. The data positions from multiple objects in the buffer can shift after merging, requiring identifying the correct data during calculations.

To fully unleash the potential of WebGPU, we introduce FusionRender, a system designed to enhance graphic rendering performance in web browsers by merging graphics rendering and reducing communication overhead. We can achieve enhanced performance by integrating established graphics rendering frameworks into FusionRender and leveraging the new renderer.

During the rendering process, FusionRender initially groups objects and consolidates the rendering of graphics within the same group, thereby reducing the frequency of CPU-to-GPU communication. To determine which objects can be merged for rendering, FusionRender analyzes each object and assigns a signature. Subsequently, signatures with the same hash value are identified, enabling the grouping of graphics based on their signatures. Graphics

within the same group are then rendered using the same rendering pipeline.

When rendering the group of objects together, the data is concatenated and transmitted to the GPU. To ensure that the data required for each graphic's rendering is accessible after merging, FusionRender employs instance indices to track individual objects. Building upon the abovementioned method, we have considered dividing the merge operations into several batches for submission.

We conducted a comprehensive performance evaluation of FusionRender. It exhibited improved rendering frame rates in the simulated experiments compared to all baseline frameworks. Compared to the best-performing baseline framework, the median performance improvement across varying scene complexities was 29.3%-122.1%. In real-world scenarios with more complex features, FusionRender delivered an overall performance boost ranging from 9.4% to 39.7%. We also examined the impact of different devices and browsers and found that FusionRender consistently excelled across diverse contexts.

While WebGPU is currently in its early implementation stage and lacks support for some features, it holds the potential for significant performance improvements beyond the baseline provided by WebGL as it evolves and matures. One notable aspect is that mainstream browsers have not yet implemented the CPU multi-threading functionality envisioned in WebGPU's design. This optimization complements our approach and can be combined with it. Our work aims to serve as an initial exploration of WebGPU's capabilities using the already supported features.

In summary, our principal contributions are as follows:

- We find the current underperformance of WebGPU-based web rendering frameworks to WebGL, indicating an underutilization of WebGPU's capabilities.
- We propose FusionRender, leveraging WebGPU's features to enhance web graphic rendering efficiency through reduced communication frequency.
- We conduct comprehensive experiments, confirming FusionRender's favorable performance optimization across diverse scenarios.

In the remaining sections of this paper, we introduce the background and motivation in section 2, discuss the system design in section 3, present the evaluations in section 4, explore related work in section 5, and conclude in section 6.

## 2 BACKGROUND AND MOTIVATION

In this section, we will initially showcase the priciples of 3D graphics rendering. We then introduce the potential of WebGPU by delving into its intended objectives and conducting simple experiments. Subsequently, we will investigate the current status of current graphics rendering frameworks in web browsers based on WebGPU.

### 2.1 Graphics Rendering on Web Browsers

Essential components required for graphics rendering are depicted in Figure 1. The renderer requires information for rendering graphics, including the camera and the scene. The camera observes the scene from a particular perspective, obtaining a two-dimensional projection of the three-dimensional scene on the plane. The scene comprises objects and light sources.

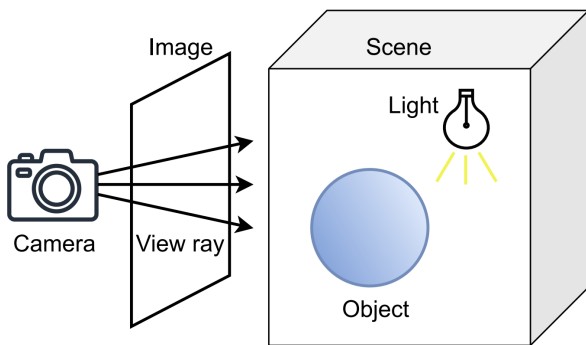

**Figure 1: Key components of a graphics rendering system**

The graphics pipeline is a sequence of operations that processes the vertices and textures of a mesh into pixels in the render target [16]. In WebGL and WebGPU, users can configure the Vertex and Fragment Stage within the rendering pipeline. In the Vertex Stage, the GPU calculates the position of each vertex, and in the Fragment Stage, the GPU computes the color of each fragment. Input data for these two stages can be configured through GPU buffers. The code that runs is called the vertex/fragment shader, and it is executed for each vertex/fragment with GPU parallel computation.

### 2.2 Potential of WebGPU

The WebGPU API [65] enables web developers to use the underlying system's GPU (Graphics Processing Unit) to carry out high-performance computations and draw complex images that can be rendered in the browser. Serving as the successor to WebGL [64], WebGPU offers enhanced compatibility with modern GPUs, allowing web applications to fully leverage GPU capabilities and attain superior performance. This is particularly advantageous for applications that demand graphics rendering capabilities.

The primary purpose of WebGPU is to address the foundational shortcomings of WebGL. While WebGL has been a widely utilized graphics API on the web since its proposal in 2011, it faces limitations in accommodating the latest hardware advancements. WebGL is rooted in the OpenGL ES API [37] and employs a global state machine. It utilizes separate APIs for configuring various parameters within the rendering pipeline. However, this approach can only partially use the capabilities of modern CPUs and GPUs, primarily due to its historical constraints. In contrast, WebGPU is constructed upon cutting-edge graphics technologies such as Direct3D 12 [52], Metal [18], and Vulkan [38].

When compared to WebGL, WebGPU offers the following advantages:

(1) **Minimized CPU Overhead:** The WebGL API closely mirrors the OpenGL ES API. On contemporary devices employing Direct3D 12, Metal, or Vulkan, notable CPU overhead stems from translation processes. In contrast, WebGPU was meticulously crafted with this concern in focus, streamlining its implementation on Metal, D3D12, and Vulkan without necessitating convoluted browser integrations. Furthermore, while WebGL/OpenGL conducts error checks for each command, resulting in substantial runtime overhead, these checks

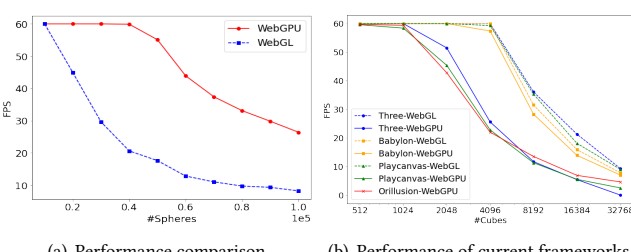

(a) Performance comparison

(b) Performance of current frameworks

**Figure 2: Performance comparison of WebGL and WebGPU (a), and performance comparison of current frameworks (b)**

are vital during development but unnecessary during runtime. WebGPU effectively sidesteps this issue entirely.

(2) **Reduced Data communication:** In WebGL, states are sequentially established, necessitating the configuration of the GPU for each state. During rendering, the GPU driver must ascertain the feasibility of global optimization. In WebGPU, the traditional approach of configuring individual states as global states has been replaced. Instead, all pipeline states are combined and effortlessly copied to the GPU's designated destination via memory copying. This approach significantly mitigates CPU-GPU communication and contributes to enhanced global optimization.

(3) **Enable CPU multithreading:** WebGPU offers better support for parallel processing with CPU multithreading. Unlike WebGL, where each command can potentially alter the global state, making it unsuitable for multithreaded parallel processing, WebGPU's execution process is divided into command recording and command execution. During command recording, all WebGPU commands are saved to a buffer, which is then submitted for execution. Since command recording is a purely CPU operation, it can be parallelized using CPU multithreading.

Due to the current lack of support for CPU multithreading in WebGPU implementations of mainstream browsers, the practical WebGPU optimization features available for graphics rendering are limited to (1) and (2) mentioned above. With the future implementation of CPU multithreading support in WebGPU, its capabilities are expected to be further enhanced.

## 2.3 Motivation Experiments

*2.3.1 Validating the Potential of WebGPU.* To practically evaluate the potential of WebGPU, we employed both WebGPU and WebGL to render an equal number of moving spheres. We made concerted efforts to maintain code consistency between the two versions beyond the realm of graphic rendering. The experimentation took place on a MacBook utilizing the Chrome browser.

Figure 2(a) visually represents the FPS (frame per second) fluctuations corresponding to the increased number of rendered spheres. It is clear that across diverse levels of graphic complexity, WebGPU-based rendering exhibits superior performance compared to WebGL.

*2.3.2 Investigating Existing WebGPU-Based Rendering Frameworks.* Despite the rapid advancement of rendering frameworks based on WebGPU, our knowledge about their performance still needs

**Table 1: Popular web-based graphics rendering frameworks**

| Framework | #github star (k) | Backend |
|---|---|---|
| **Three.js [14]** | 93.9 | WebGL/WebGPU |
| **Babylon.js [3]** | 21.2 | WebGL/WebGPU |
| **Playcanvas [11]** | 8.6 | WebGL/WebGPU |
| **Orillusion [9]** | 2.2 | WebGPU |

to be improved. Therefore, we want to understand the current performance of WebGPU-based graphics rendering frameworks.

Table 1 presents the most popular WebGPU-based graphics rendering frameworks at present. We searched GitHub using the keywords WebGPU, selecting Web graphics rendering frameworks with more than 1,000 stars. We proceeded to compare WebGPU-based graphics rendering frameworks with their WebGL counterparts.

Initial performance tests were conducted by rendering varying numbers of cubes on a MacBook using the Chrome browser. The results are depicted in Figure 2(b). Surprisingly, we discovered that the current state of WebGPU-based graphics rendering is inferior to that of WebGL-based. This misalignment with the potential of WebGPU is unexpected.

Having investigated both existing frameworks and the features of WebGPU, we have identified that the primary factor behind this issue is the lack of proper alignment between current frameworks and the specific attributes of WebGPU. Existing frameworks process different graphics separately, resulting in the separate transmission of data associated with each graphic to the GPU. This approach fails to take advantage of WebGPU's ability to reduce communication between the CPU and GPU. Due to WebGL's use of a global state machine, sequential configuration of different graphic states is required, whereas WebGPU offers a more optimal implementation approach.

## 3 FUSIONRENDER

In this section, we will begin by offering insights into the design of our system. Next, we will provide an overview of the system, followed by an introduction to its key components, including object grouping and merged rendering.

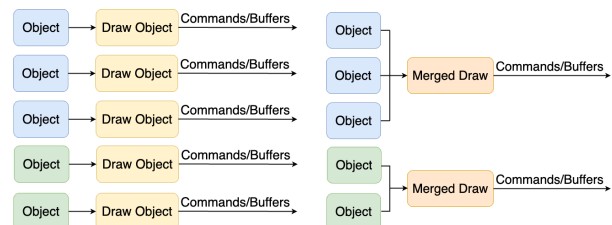

**Figure 3: Merging rendering processes of distinct objects**

## 3.1 Design Insights

In order to enhance the performance of web-based graphics rendering, it is imperative to leverage the capabilities of WebGPU fully. It is crucial to merge the rendering processes of distinct objects whenever possible to reduce redundant communication. The process of object consolidation is illustrated in Figure 3. Current

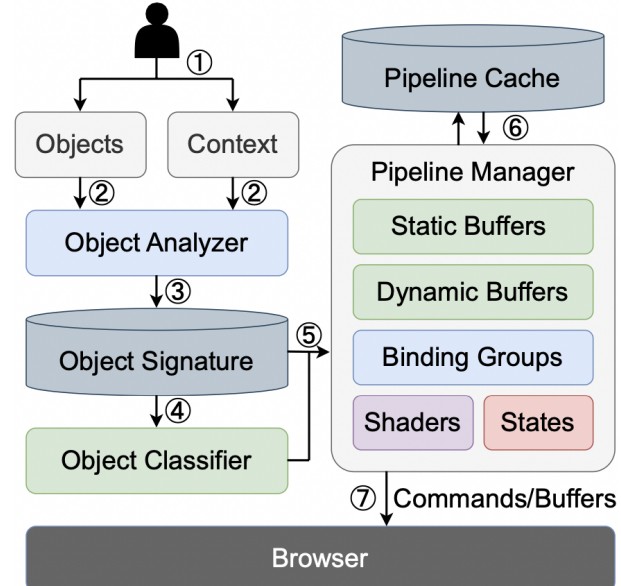

**Figure 4: Overview of FusionRender**

frameworks employ the approach depicted on the left side of Figure 3, where each object is rendered separately, leading to individual transmission of drawing commands and graphic data buffers to the GPU. The desired system, as illustrated on the right side of Figure 3, involves merging the rendering of different objects to the greatest extent possible. After grouping the objects, each group can be rendered collectively, and the drawing commands and data for the entire group are transmitted to the GPU as a unified unit.

### 3.2 Overview of FusionRender

To achieve our goals and tackle challenges, we introduce FusionRender, which initiates by analyzing the attributes of graphics and subsequently endeavors to consolidate graphic rendering based on the analysis outcomes.

Figure 4 provides an overview of FusionRender. The system takes user-defined configurations for canvas context and objects as input and generates WebGPU commands and buffers as output. When integrated with existing rendering frameworks, users can define the content they want to render using the APIs of those frameworks. FusionRender then extracts the required information from this content, effectively taking over the rendering process and the output generation initially handled by the framework's renderer.

Firstly, FusionRender extracts the necessary objects and context information (①). During scene rendering, the renderer initially forwards these configurations to the object analyzer for examination (②), intending to acquire pertinent information for configuring the rendering pipeline. The object analyzer obtains a distinctive "signature" for each object (③), encompassing all requisite data for configuring the rendering pipeline. Objects with matching signatures can be rendered using a single rendering pipeline. The object

classifier then segregates objects into groups based on their signatures (④), allowing objects within the same group to be drawn using a shared rendering pipeline.

Following this, for each object cluster, a dedicated rendering pipeline is crafted and employed by their signatures and grouping outcomes (⑤). The essential requisites for configuring the rendering pipeline include shaders, static buffers, dynamic buffers, bindgroups, and other states. Shaders encompass code intended for GPU execution. Static buffers house almost unvarying data throughout the rendering process, often including geometric information, while dynamic buffers accommodate data that may fluctuate during rendering, such as transformations for graphics. Binding groups elucidate how data from dynamic buffers is conveyed to designated variables within the shader code. Other states encompass additional configuration options, including depth-stencil settings, multisampling, and primitives, with these configurations being numerical, boolean, or enumerated values.

A pipeline cache is implemented to circumvent the creation of redundant pipeline and GPU buffer, verifying content availability during each frame's rendering (⑥). The requisite data and commands for drawing each set of objects are submitted to the browser, culminating in GPU-driven rendering (⑦). Thus, FusionRender optimizes by minimizing communication interactions with the GPU, culminating in the efficient realization of web-based 3D graphic rendering.

### 3.3 Object Grouping

To determine which objects can be combined for rendering, FusionRender groups them. Objects within the same group can be rendered using a single rendering pipeline. Algorithm 1 provides pseudocode for this grouping process. Given a canvas context and a collection of objects defined by the user as set $S$, the goal is to obtain a list of grouped objects $G[]$, where each element in $G$ is a list of objects.

To decide which objects should belong to the same group, we give a signature to each object and group objects with identical signatures together. The $getObjectSignature$ function is employed to obtain the signature of each object (Line 4). To identify objects with matching signatures, we establish a map $M$ (Line 1). Its keys are the hash values of the signatures, and the corresponding values are lists of objects sharing that signature . A signature consists of a series of names and values representing the configuration requirements of the object's rendering pipeline. The hash value of a signature is obtained by concatenating the names and values and converting them into a string (Line 6). During the grouping process, each object's signature hash value is checked in $M$. If it already exists, the object will be added to the corresponding list. Otherwise, a new entry will be created in $M$ (Lines 7-12).

To describe the required rendering pipeline configuration for an object, it is necessary to specify the contents of several components in the signature: attributes, shaders, fragment target, primitive, depth-stencil, and multisample. Here's a breakdown of each component:

- **Attributes**: representing the expected layout of static data used in the vertex shader stage
- **Shaders**: describing the shaders used in the pipeline

---

**Algorithm 1** Grouping Objects by Signature

---

**Input**: A set of objects: $S = \{object_1, object_2...object_n\}$, Canvas context: $context$

**Output**: A list of grouped object lists: $G[]$

1: INITIAL $M \leftarrow \{\}$           ▷ initial M to empty map
2: INITIAL $G \leftarrow []$           ▷ initial G to empty list
3: **for** $object_i$ in $S$ **do**
4:     $signature \leftarrow getObjectSignature(object_i, context)$
5:     $object.signature \leftarrow signature$
6:     $hashKey \leftarrow hashSignature(signature)$
7:     **if** $hashKey$ exists in $M$ **then**
8:        add $object_i$ to $M[hashKey]$
9:     **else**
10:        $objectList \leftarrow []$
11:        add $object_i$ to $objectList$
12:        add $< hashkey, objectList >$ to $M$
13:     **end if**
14: **end for**
15: **for** $(hashKey, objectList)$ in $M$ **do**
16:     add $objectList$ to $G$
17: **end for**
18: **return** $G$

---

- **Fragment Target**: describing color states that provide configuration details for the colors output by the fragment shader stage
- **Primitives**: describing how a pipeline constructs and rasterizes primitives from its vertex inputs
- **Depth-stencil**: including both the pipeline's depth properties and stencil properties
- **Multisample**: describing how the pipeline interacts with a render pass's multi-sampled attachments

Table 2 displays the names and data types of specific values for each component. In the table, "T" means type, "N" represents a number, "E" indicates an enumerated type, "B" signifies a boolean, "F" denotes bitwise flags, and "M" represents a bitmask. While the number of attributes may vary related to the number of static data buffers used during rendering, all other entries only appear once in the signature.

## 3.4 Merged Rendering

After classifying objects based on their signatures, merged rendering can be performed for each group. Algorithm 2 provides pseudocode for merged rendering for the first frame. Given the global scene ($S$), camera ($C$), the group of objects needed to be rendered ($G_k$), and the $device$ abstraction provided by WebGPU, the goal is to commit the rendering commands and data buffers to the browser, ultimately completing GPU calculations and graphical display. Several steps are involved during this process, including pipeline configuration, dynamic buffer configuration, binding group configuration, static buffer configuration, and drawing.

Pipeline creation and configuration can be accomplished using the signature, which includes all the information required to create and configure the rendering pipeline (Lines 1-3). During the pipeline construction process, the layout of static attributes is generated based on the information in the "attributes" part of the signature

**Table 2: Variable names and types in the signaure**

| Component | Name | T | Name | T |
|---|---|---|---|---|
| **Attributes** | arrayStride | N | slot | N |
| | format | E | offset | N |
| | stepMode | E | — | — |
| **Shaders** | type | E | — | — |
| **Fragment Target** | colorFormat | E | alphaBlendSrc | E |
| | alphaBlendDst | E | alphaBlendOp | E |
| | colorBlendSrc | E | colorBlendDst | E |
| | colorWriteMask | E | colorBlendOp | F |
| **Primitives** | stripIndexFormat | E | topology | E |
| | frontFace | E | cullMode | E |
| **Depth-stencil** | depthStencilFormat | E | depthCompare | E |
| | depthWriteEnabled | B | stencilCompare | E |
| | stencilDepthFailOp | E | stencilFailOp | E |
| | stencilReadMask | M | stencilPassOp | E |
| | stencilWriteMask | M | — | — |
| **Multisample** | sampleCount | N | — | — |

---

**Algorithm 2** Merged Rendering of a group of objects

---

**Input**: Scene: $S$, Camera: $C$, A group of objects $G_k$, Device: $D$

**Output**: commands and data buffers committed to browser

1: $signature \leftarrow G_k[0].signature$
2: $pipeline \leftarrow createRenderPipelineBySignature(signature)$
3: $device.setPipeline(pipeline)$      ▷ Pipeline
4: $mergedArrays \leftarrow []$
5: **for** $object_i$ in $G_k$ **do**
6:     $dynamicAttributes \leftarrow getDynamic(S, C, object_i)$
7:     add $dynamicAttributes$ to $mergedArrays$
8: **end for**
9: $bindGroup \leftarrow createBindgroup(mergedArrays)$
10: $device.writeBuffers(mergedArrays)$    ▷ Dynamic Buffers
11: $device.setBindGroup(bindGroup)$     ▷ Binding Groups
12: $mergedArrays \leftarrow []$
13: $staticOffsets \leftarrow []$
14: $offsets \leftarrow 0$
15: **for** $object_i$ in $G_k$ **do**
16:     $staticAttributes \leftarrow getStatic(object_i)$
17:     add $staticAttributes$ to $mergedArrays$
18:     $offsets \leftarrow offsets + staticAttributes.lengths$
19:     add $offsets$ to $staticOffsets$
20: **end for**
21: $device.writeBuffers(mergedArrays)$     ▷ Static Buffers
22: **for** $i$ **from** $0$ to $G_k.length$ **do**
23:     $count \leftarrow G_k[i].vertexCount$
24:     $offset \leftarrow staticOffset[i]$
25:     $device.draw(count, offset, i)$      ▷ Draw
26: **end for**

---

to build descriptors. The vertex and fragment shader codes are dynamically composed based on shader type and lighting conditions. Additionally, various other states are appropriately placed within the pipeline descriptor.

Dynamic buffers contain data that needs to be updated every frame during rendering, including camera viewpoints, scene angles, and graphics properties like position, scale, rotation, color, opacity,

and more. In the past, multiple variable data would be concatenated and passed into shaders as a *struct* data structure for a single object. For multiple objects, data from individual objects is concated based on a single object's dynamic data and passed into shaders as a *struct array* for interpretation (Lines 4-8). The binding group presents the layout of dynamic attributes and defines how these resources are used in shader stages (Lines 9-11).

Static buffers, conversely, contain data that remains nearly constant during rendering and typically correspond to a set of data for each vertex of the graphics, including vertex attributes such as position, normal, and UV coordinates. During merged rendering, static data from different graphics is concatenated, and the offset of each graphic's data in the large array is recorded (Lines 12-21).

Finally, graphics rendering is performed through the vertex and fragment stages (Lines 22-26). Each vertex's static attributes are passed directly as arguments to the vertex shader's 'main' function in the vertex stage. WebGPU supports reading data from static buffers starting from a specified offset. By passing vertex counts and static offsets to the draw function, we can retrieve the corresponding static data for each graphic. Furthermore, the graphic's index is passed to the draw function, allowing us to locate the specific data from the merged dynamic data by finding the needed *struct* from the *struct array*. For the vertex stage, the index can be obtained using WebGPU's built-in functions. The index can be included in the output of the vertex stage and subsequently used as input in the fragment stage, completing the index transfer process.

**Grid Search.** We can consider rendering multiple small groups instead of one large group during the merging process. When constructing merged data, the GPU idles, creating a GPU bubble. Splitting the process into several smaller groups allows the GPU to handle some of the content while the CPU prepares data for subsequent rendering. However, merging into smaller groups introduces more CPU overhead and data transfer operations than merging into a large group, leading to a tradeoff in this approach.

**Pipeline Cache.** In rendering subsequent frames, we employ caching to prevent the redundant creation of pipelines, bind groups, and data buffers. We can retrieve them from the cache and configure them in their appropriate positions within the render pass.

## 4 EVALUATIONS

In this section, we will start by introducing the implementation and experimental setup. Subsequently, we will present the results of simulated experiments and analyze the impact of various browsers and devices. Finally, we will discuss the experimental results in real-world cases.

### 4.1 Implementation

We implement a prototype of FusionRender for Three.js. We use our renderer for the WebGPU rendering process. For the shared aspects of graphics rendering, such as classes representing three-dimensional graphics, cameras, and lighting, we integrated code from three.js [14]. We can apply a similar integration approach to replace the renderer and seamlessly incorporate it with our system for other frameworks, as most graphics rendering frameworks include the key components illustrated in Figure 1 [3, 9, 11, 14].

Our implementation comprises over 4k lines of JavaScript and WGSL [66] code, encompassing both graphics grouping and the utilization of WebGPU for graphics rendering[1]. For a set of graphics that can be rendered using the same rendering pipeline, we conducted a grid search. We separately attempted to combine the data into 1, 2, 3, and 4 groups using WebGPU storage buffers and explored having a sufficient number of groups to use uniform buffers, ultimately selecting the best-performing configuration. The WebGPU uniform buffer can hold less data than the storage buffer, but it offers faster speed.

### 4.2 Evaluation Setup

The experimental equipment consists of a MacBook Pro (with Apple M1, MacOS 12, and 16 GB RAM), a ThinkPad X1 Yoga (with Intel i5, Windows 11, and 8 GB RAM), and a Pixel 6 (with Google Tensor, Android 12, and 8 GB RAM). The browsers used for the experiments include Chrome Dev (version 120.0.6051.2) and Firefox Nightly (version 120.0a1). We utilized MacBook Pro and the Chrome browser to explore the overall system performance and discussed the impact of devices and browsers with different configurations.

Our experiments are divided into simulated experiments and real-world scenario experiments. In the simulated experiments, we progressively increase the number of rendered objects and compare the performance of FusionRender with other baselines. In the real-world scenario experiments, we use real cases of the three.js framework and compare the rendering performance of FusionRender with three.js in these real-world scenarios. When measuring performance, we wait for the graphics to load first and then measure the average FPS for the following 1 minute.

### 4.3 Simulated Evaluation

*4.3.1 Overall Performance.* Previous studies have indicated that the number of graphics influences visual realism, and participants exposed to a higher degree of visual realism experience a stronger sense of presence [33, 69]. More complex scenes can enhance user engagement [62] and enable more accurate judgments [49], and when training with graphics software, approaching real-world complexity yields better results [56].

Therefore, we conduct simulated experiments to test the performance of FusionRender under varying levels of scene complexity. We progressively increase the number of rendered cubes ([512, 1024, 2048, 4096, 8192, 16384, 32768]) and measure the performance of FusionRender as well as existing frameworks as shown in Table 1. The experimental results using the Chrome browser on a MacBook Pro are depicted in the leftmost subgraph labeled "Chrome-MacBook" in Figure 5. Under different levels of scene complexity, FusionRender outperforms existing frameworks.

As performance improvements vary across different levels of scene complexity, we report our experimental results using the median performance enhancement. We exclude scenarios where both baseline frameworks perform smoothly (>58FPS) due to their relative simplicity. In the remaining scenarios, we calculate the median enhancement to represent the performance improvement of FusionRender compared to a specific baseline. Compared to the previously top-performing baseline, Three.js-WebGL, FusionRender

---

[1]The code and data will be open-sourced later

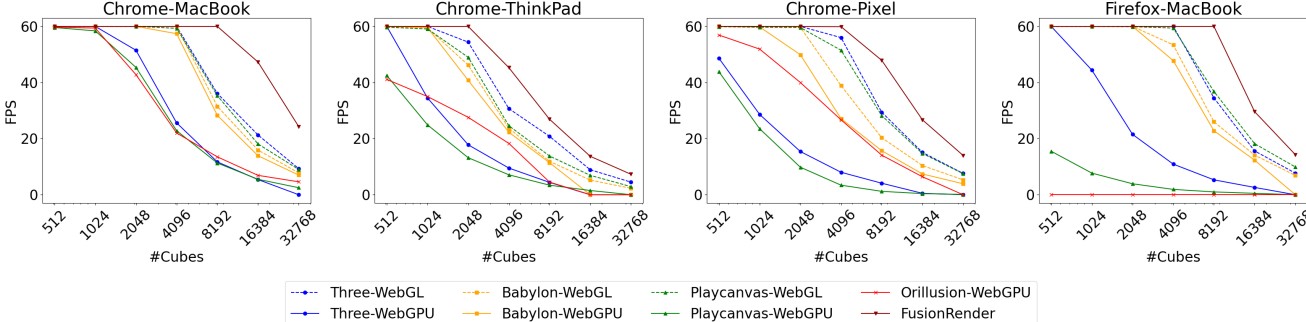

**Figure 5: Performance of different frameworks in simulated evaluation under various scenarios**

achieves a median performance enhancement of 122.1%. Furthermore, compared to Playcanvas-WebGPU, FusionRender's median performance enhancement can reach as high as 431.0%.

*4.3.2 Impact of Heterogeneity.* We investigated the impact of different devices and browsers on FusionRender, and the results are presented in Figure 5.

For different devices, we consistently used the Chrome browser and conducted experiments on a MacBook Pro, a ThinkPad X1 Yoga, which runs different operating systems (MacOS and Windows, respectively), and a Pixel 6 smartphone with the Android OS. From the subgraphs "Chrome-MacBook," "Chrome-ThinkPad," and "Chrome-Pixel" in Figure 5, it is evident that FusionRender consistently exhibits performance improvements compared to the baseline on various devices. The median performance enhancements of FusionRender compared to the best baseline are 29.3% for the ThinkPad laptop and 75.7% for the Pixel smartphone.

For different browsers, we maintained the use of a MacBook Pro and switched between the Chrome and Firefox browsers. The subgraphs "Chrome-MacBook" and "Firefox-MacBook" in Figure 5 illustrate the experimental results. It can be observed that Fusion-Render outperforms the baseline in both browsers. FusionRender achieves a median performance enhancement of 62.6% compared to the best baseline in the Firefox browser.

## 4.4 Real Case Study

In order to explore the performance of FusionRender in real-world scenarios with more advanced functionality and complex scenes, we compared FusionRender, Three.js-WebGL, and Three.js-WebGPU. We selected examples from the Three.js Forum [6] posted last year, focusing on open-source examples with performance issues on mobile devices and excluding those with custom GLSL shaders [8].

The cases used are illustrated in Figure 7 and include the following three:

- **PinusTree [10]**: Rendering a pine tree composed of various graphical elements with multiple hierarchical levels, including more complex lighting and materials.
- **ForceGraph [1]**: Drawing force-directed graphs where the distance between points and the magnitude of forces between points are related.
- **BubblePose [4]**: Identifying human body positions and rendering them using bubble representations.

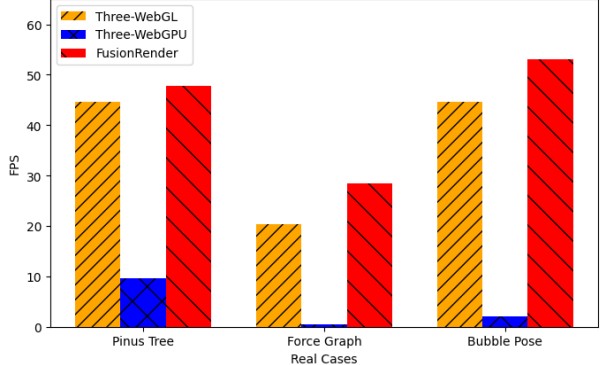

**Figure 6: Performance comparison of real cases**

**Table 3: Effects of grid search on performance (FPS)**

|            | WebGL | S-1  | S-2  | S-3  | S-4  | U    |
|------------|-------|------|------|------|------|------|
| **Simulated**  | 7.6   | **14.0** | 13.9 | 13.6 | 13.6 | 9.9  |
| **PinusTree**  | 44.6  | 35.9 | 45   | 46.3 | 44.5 | **47.7** |
| **ForceGraph** | 20.4  | **28.5** | 25.0 | 13.7 | 11.3 | 1.4  |
| **BubblePose** | 44.6  | 44.3 | **53.0** | 49.7 | 49.6 | 34.4 |

These examples were initially designed for Three.js-WebGL. We made minor adjustments to make them compatible with Three.js-WebGPU and FusionRender. Moreover, we remove components for neural network inference and focus solely on rendering. Throughout this modification process, we ensured the examples remained consistent across the three comparative frameworks.

The evaluation was conducted on a Pixel 6 smartphone with Chrome browser, and the results are depicted in Figure 6. As we can see, FusionRender consistently outperformed Three.js-WebGL in various real-world scenarios. Specifically, for PinusTree, Force-Graph, and BubblePose, FusionRender demonstrated performance improvements of 9.4%, 39.7%, and 18.8%, respectively. Due to the need for calculating hierarchical graphics positions or greater GPU computational demands in real-world scenarios, the performance improvements are minor than in simulated experiments.

## 4.5 Impact of Grid Search

We analyzed the impact of the grid search on performance. These experiments were conducted on a Pixel smartphone using the Chrome

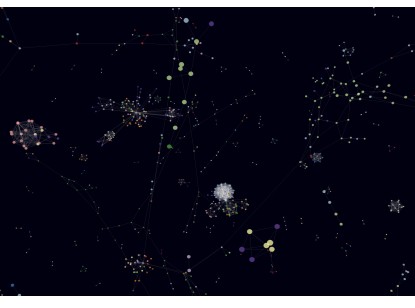

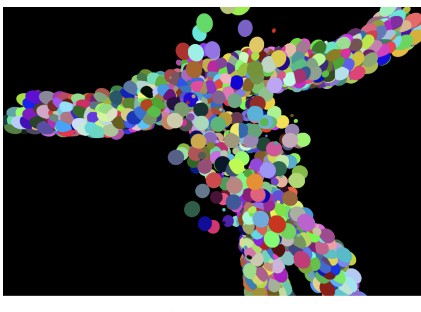

| (a) PinusTree | (b) ForceGraph | (c) BubblePose |

**Figure 7: The application scenarios employed in the real case study**

browser, and the results are presented in Table 3. In the table, 'We-bGL' represents the results obtained with Three.js-WebGL. Additionally, 'S-1', 'S-2', 'S-3', and 'S-4' refer to scenarios in which graphics that could be rendered using a single pipeline were divided into 1, 2, 3, and 4 groups, respectively, and submitted using storage buffers. 'U' denotes scenarios where the data was divided into a sufficient number of submissions to utilize uniform buffers. The table displays the results for rendering 32,768 cubes for the simulated experiments.

The optimal choice varies for different scenarios. In cases with relatively few overall graphics, such as 'Pinus Tree,' the use of uniform buffers yields better results, while for other scenarios, the finer grouping overhead offsets the benefits of uniform buffers. In the case of 'BubblePose,' submitting one group for GPU calculation while simultaneously merging the next set of data can reduce GPU 'bubble,' resulting in more significant performance gains that outweigh the additional overhead.

## 5 RELATED WORK

**Rendering Optimization** Due to the limited resources of devices and the high computational demands of 3D rendering, significant research has been devoted to rendering optimization. Many studies propose efficient rendering frameworks for modern hardware [20, 41, 60, 74], optimizing power consumption [30, 45], performance models [29], and memory usage [36]. Some also focus on ray tracing [57], point cloud rendering optimization [59], and utilize parallel computing and load balancing for VR rendering [40, 47, 48, 55]. However, these efforts primarily address native 3D rendering and have yet to specifically explore the unique challenges of rendering in web browsers.

**Web3D** Web3D serves as the foundation for many web applications. Numerous studies have been conducted to explore the diverse applications of Web3D in various fields, including efficient display of geographical information [35, 46], weather monitoring [61], visualization of proteins and molecular structures [23, 71], and art exhibitions [72], etc. Additionally, optimizing Web3D has also garnered attention. Some research has focused on measuring and analyzing the performance of existing frameworks [22], while others have investigated methods to reduce model size and achieve more efficient rendering of large models [32, 75]. Furthermore, specific features of Web3D have been the subject of research. For instance,

studies have delved into presenting 3D point clouds using data space and hierarchical details [24], large-scale 3D dataset retrieval techniques [28], custom animation creation [27], and leveraging interactive capabilities between 3D and 2D elements to achieve desired objectives [68]. Despite these explorations into various aspects of Web3D, research has yet to optimize three-dimensional graphics rendering within web browsers using WebGPU.

**WebGL/WebGPU** Some research focuses on WebGPU's shader language, WGSL [66], and compiler. WGSLsmith [53] provides a testing toolkit for testing WGSL compilers through randomized testing. Google [25] reported their experience using fuzzing testing to discover errors in WGSL compilers. Additionally, MC Mutants [43] and GPUHarbor [42] researched memory consistency specification testing for WebGPU. Furthermore, some work focuses on the performance optimization capabilities of WebGPU. For instance, WebDNN [31] utilizes WebGPU to accelerate browser-based deep neural network inference, and GraphWaGu [26] optimizes web-based graph visualization by WebGPU. On the other hand, some research focuses on WebGPU's predecessor, WebGL. Several works have implemented applications in various domains based on WebGL, including medicine [39, 44], archaeology [63], and biology [58]. Other works have studied the security issues of WebGL ( Milkomeda [73], UNIGL [70], and GLeeFuzz [54]). Our work explores the untapped potential of utilizing WebGPU for enhanced rendering optimization, an area that earlier studies have yet to address.

## 6 CONCLUSION

In this paper, we introduce FusionRender, a system designed to comprehensively leverage the capabilities of the next-generation web graphics API, WebGPU, for graphics rendering in web browsers. Compared to previous graphics rendering frameworks based on WebGL or WebGPU, FusionRender optimizes performance by efficiently merging draw calls, thus reducing redundant communication. In simulated experiments and real-world scenarios, as well as across different devices and browsers, FusionRender has demonstrated notable improvements. Further enhancements are currently in progress. Firstly, our research will focus on balancing energy efficiency, performance, and graphical quality enhancements. Secondly, we will address optimization challenges when both the WebGPU general compute and rendering pipelines are active.

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
