# OpenReview forum: "FusionRender: Harnessing WebGPU’s Power for Enhanced Graphics Performance on Web Browsers"
_ACM.org/TheWebConf/2024/Conference — TheWebConf24 Oral_

### Official Review · Reviewer_ovUs · 2023-11-07

**Novelty:** 4
**Technical Quality:** 6

**Review:**

This paper introduced a system called FusionRender to enhance the performance of WebGPU-backended graphics rendering frameworks. After analyzing the reasons for the poor performance of the current WebGPU-backended graphics rendering framework, this paper proposed two steps including object grouping and merged rendering to improve performance.

Strength:
1. Clear motivation and optimization goal.
2. Well-structured and good writing.

Weakness:
1. The contribution is limited. This work only contributes to the rendering framework and finished the performance enhancement which should be done by the frameworks themselves.
2. The method is not novel and a common idea. The paper didn't provide the details of the key point of the whole algorithm. How the signature is calculated is the key which influences the grouping results significantly.

**Questions:**

1. How the system `getObjectSignature`?
2. As the method mainly improved the performance by grouping objects with the same signature, is the simulation experiment which tested by increasing the number of the same objects convincing? How about increasing object numbers with different signatures?

**Reviewer Confidence:**

3: The reviewer is confident but not certain that the evaluation is correct

**Scope:**

3: The work is somewhat relevant to the Web and to the track, and is of narrow interest to a sub-community

---

### Official Review · Reviewer_i17X · 2023-11-14

**Novelty:** 4
**Technical Quality:** 4

**Review:**

This paper addresses the challenges associated with web graphic rendering, particularly the underutilization of the next-generation web graphics API, WebGPU. The authors propose a system, FusionRender, aimed at maximizing the efficiency of graphics rendering by reducing redundant communication through the consolidation of draw calls. The performance of the approach is evaluated across various scenarios, devices, and browsers, providing valuable insights into the potential enhancements offered by WebGPU.

Strengths:
•	The investigation of the underperformance of current WebGPU-based frameworks compared to WebGL provide valuable insights into the existing gaps and the need for optimization.
•	The proposed approach to maximize communication efficiency by consolidating draw calls is innovative. The system's analysis of objects, assignment of signatures, and subsequent grouping for rendering showcases a thoughtful design addressing the identified challenges.
•	FusionRender achieves promising results in both simulation and real-world experiments.

Weaknesses:
•	The authors should discuss the generalizability of proposed framework to different types of web applications beyond the ones mentioned in the paper. This would provide a broader perspective on the potential impact of the proposed approach.
•	The paper mentions preliminary support in Chrome and Firefox, but the impact of varying levels of support in other browsers (especially those with limited or no WebGPU support) could be discussed. Addressing the dependency on browser adoption and potential challenges in achieving widespread compatibility would provide a more holistic view.
•	The paper focuses on performance improvements across different scenarios, but it would be beneficial to delve into the scalability of FusionRender. A discussion on the scalability limitations and potential strategies for handling more intricate graphics scenarios could enhance the paper's practical insights.

I acknowledge that I have read the rebuttal.

**Questions:**

•	How well does FusionRender handle increasingly complex scenes or larger datasets?
•	Energy efficiency considerations are crucial, especially in the context of web applications running on diverse devices.  It would be worthwhile to discuss the framework’s trade-offs in balancing energy efficiency and graphical quality.
•	Add a discussion on the current limitations of the framework and the future extensions possible.

**Ethics Review Description:**

No ethical issues in the paper

**Reviewer Confidence:**

1: The reviewer's evaluation is an educated guess

**Scope:**

3: The work is somewhat relevant to the Web and to the track, and is of narrow interest to a sub-community

---

### Official Review · Reviewer_ibGq · 2023-11-18

**Novelty:** 3
**Technical Quality:** 4

**Review:**

This study focuses on the difficulties related to web visual rendering, specifically the insufficient utilisation of the advanced web graphics API, WebGPU.   This study suggests two measures, namely object grouping and merged rendering, to enhance the performance of the current WebGPU-backed graphics rendering framework after analysing the causes of its poor performance.  Nonethess, the issues are also well-known among the researchers or industry partners.

Then, the authors present FusionRender, a technology designed to optimise graphics rendering by minimising unnecessary communication through the consolidation of draw calls.   The approach's performance is assessed in several scenarios, devices, and browsers, yielding significant insights into the possible improvements afforded by WebGPU.



Key Strengths: The article contains evident movitation and objective for optimisation of web-based metaverse/graphics. Also, the writing is well-organized and of high quality.   An examination of the subpar performance of current WebGPU-based frameworks in comparison to WebGL yields significant insights into the existing deficiencies and the necessity for optimisation.   The suggested strategy to optimise communication efficiency through the consolidation of draw calls is groundbreaking.   The system's analysis of objects, allocation of signatures, and subsequent categorization for display demonstrates a well-considered design that effectively tackles the highlighted difficulties.   FusionRender demonstrates favourable outcomes in both simulated and real-world trials.

Major Pitfall: The authors should address the applicability of the suggested framework to various sorts of web applications not explicitly covered in the paper.   This would offer a more comprehensive outlook on the potential ramifications of the suggested method.   The paper acknowledges initial endorsement in Chrome and Firefox, but it fails to address the potential ramifications of divergent levels of support in other browsers, particularly those with restricted or nonexistent WebGPU compatibility.   The contribution is restricted.   This work solely contributes to the rendering framework and completes the performance boost that should be carried out by the frameworks themselves.

However, the approach is not innovative and is a widely known concept.   The study lacked specific information regarding the pivotal aspect of the entire method.   The calculation of the signature is a crucial factor that has a considerable impact on the grouping outcomes.
To have a comprehensive perspective, it is necessary to consider the reliance on browser adoption and the potential obstacles to attaining broad compatibility.   The research primarily addresses enhancements in performance across many contexts; nevertheless, it would be advantageous to explore the scalability of FusionRender further.   Incorporating a discourse on the constraints of scalability and various approaches for managing complex graphics scenarios could augment the paper's pragmatic observations.

**Questions:**

For the metaverse, the major and most critical issue is the running and operating costs of the server, which lead to financial burden and energy costs. Could the author elaborate more on how the work can reflect on the aforementioned issues?

**Ethics Review Description:**

Nil

**Reviewer Confidence:**

3: The reviewer is confident but not certain that the evaluation is correct

**Scope:**

3: The work is somewhat relevant to the Web and to the track, and is of narrow interest to a sub-community

---

### Official Review · Reviewer_t5X8 · 2023-11-23

**Novelty:** 4
**Technical Quality:** 5

**Review:**

The legacy WebGL web graphics API has its limitations, which push the debut of WebGPU. However, evaluations show that WebGPU-based rendering frameworks currently still underperform WebGL likely due to the underutilization of its full potential. In this paper, the authors aim to tackle the problem by designing FusionRender, a framework for optimizing graphics rendering on web browsers by utilizing the capabilities of WebGPU. FusionRender groups objects based on unique signatures and consolidates their rendering to minimize redundant CPU-GPU communication. This approach led to a significant improvement over existing works in rendering performance across various devices and browsers, in both simulated and real-world experiments. The authors take a first step to explore the potential of WebGPU, more effectively leveraging WebGPU's features.

Meanwhile, I have also found a couple of aspects where the authors can further enhance to improve this study. The paper does not fully address the practical challenges of implementing and integrating FusionRender into existing systems. While I understand WebGPU is in its early stage and lacks some features, it would be great to have some discussion on this, especially because the effectiveness of FusionRender is tied to the features and support of WebGPU.

The paper primarily focuses on specific use cases and configurations, which may not cover the entire spectrum of web applications and demonstrate its generalizability.

I am also curious about some broader implications of its findings, for example, in how FusionRender might influence future web application development and user experience. I look forward to seeing some explanations in the paper.

**Questions:**

How does FusionRender integrate with existing web graphics rendering frameworks? What could be the challenges involved?

As WebGPU is still evolving, how does FusionRender plan to adapt to future changes and new features in WebGPU?

Can you provide some insights into how FusionRender's performance improvements might translate to a broader range of web applications beyond the tested scenarios?

**Reviewer Confidence:**

2: The reviewer is willing to defend the evaluation, but it is likely that the reviewer did not understand parts of the paper

**Scope:**

4: The work is relevant to the Web and to the track, and is of broad interest to the community

---

### Official Review · Reviewer_XykM · 2023-11-28

**Novelty:** 4
**Technical Quality:** 5

**Review:**

Summary:

The paper introduces FusionRender, a system that is intended to comprehensively exploit the capabilities of the next-generation web graphics API, WebGPU, for graphics rendering in web browsers. Compared to previous graphics rendering frameworks that were based on WebGL or WebGPU, FusionRender optimizes performance by merging draw calls, thus greatly reducing redundant communication. In both simulated experiments and real-world scenarios, as well as across various devices and browsers, FusionRender has demonstrated improvements.

Strength:

+ Strong evaluation results

The improvement in the experiment is significant for both simulated and real-world examples.

+ Actionable insights

The improvement calls for more support on optimizations to WebGPU.

Weakness:

- Capped performance

It is unclear that the authors capped the performance at 60 FPS, especially for # Cubes for Firefox-MacBook in Figure 5. Five of seven plots are capped; therefore, the benefits of FusionRender are not fully presented. I would suggest removing the cap and adding a line of 60 FPS, as it is the suggested performance threshold, or changing Figure 5 to rendering time per frame (50 FPS means 20 msec per frame).

- Unexplained experiment results

The poor performance of Three.js’s WebGPU version in Section 4.4 is not explained. I assume these examples are not optimized for WebGPU as “these examples were initially designed for Three.js-WebGL.” I would suggest removing these data as these are never explained and will distract the user from comparing WebGL and FusionReader.

- Unclear real-world examples

The authors should summarize the challenge level of these real-world examples compared with simulated ones, like how many rendering units are included in each example.

**Questions:**

1. What are the GPU models used on the three platforms?

2. Could you provide additional details on the real-world examples, with a focus on highlighting their differences compared to the simulated instances?

3. What accounts for the poor performance of Three.js's WebGPU version in Section 4.4?

**Reviewer Confidence:**

2: The reviewer is willing to defend the evaluation, but it is likely that the reviewer did not understand parts of the paper

**Scope:**

4: The work is relevant to the Web and to the track, and is of broad interest to the community

---

### Decision · Program_Chairs · 2024-01-22

**Decision:**

Accept (Oral)

**Comment:**

This paper investigates the WebGPU framework for GPU support in web browsers and proposes FusionRender that is able to notably improve rendering performance across a broad range of typical web applications. This is effectively achieved by reducing the communication overhead between CPU and GPU for which a signature-based scheme is developed.

 The reviewers raised a number of points that were all well answered to their satisfaction (as far as they commented or the area chair could infer). Especially, questions about application complexity in the evaluation, the range of supported applications, and the underlying GPUs used and some performance presentation-related questions could be clarified. The resulting additional insights are easy to add to a final version of the paper. The authors also responded to browser support, currently only available for the two main engines from Google Chrome and Mozilla Firefox, again to be included in the paper text.

 The only open point remained the (impact on) energy consumption, which is credibly beyond the scope of this paper and subject to ongoing work of the authors.

 Overall, the paper appears spot on for the track and the conference and received great support by the reviewers. Hence my accept.